# Blood Cell Ratios Unveiled: Predictive Markers of Myocardial Infarction Prognosis

**DOI:** 10.3390/healthcare12080824

**Published:** 2024-04-13

**Authors:** Cosmina Elena Jercălău, Cătălina Liliana Andrei, Roxana Oana Darabont, Suzana Guberna, Arina Maria Staicu, Cătălin Teodor Rusu, Octavian Ceban, Crina Julieta Sinescu

**Affiliations:** 1Department of Cardiology, “Bagdasar Arseni” Emergency Hospital, University of Medicine and Pharmacy “Carol Davila”, 011241 Bucharest, Romania; roxana.darabont@umfcd.ro (R.O.D.); crina.sinescu@umfcd.ro (C.J.S.); 2Department of Cardiology, Emergency Hospital “Bagdasar-Arseni”, 050474 Bucharest, Romania; suzana.guberna@umfcd.ro (S.G.); arina-maria.staicu@rez.umfcd.ro (A.M.S.); 3Department of Internal Medicine, “Coltea” Clinical Hospital, 030167 Bucharest, Romania; catalin-teodor.rusu@rez.umfcd.ro; 4Economic Cybernetics and Informatics Department, The Bucharest University of Economic Studies, 010374 Bucharest, Romania; octavianceban1995@gmail.com

**Keywords:** cardiovascular disease, biomarker, risk stratification, prognosis, therapy

## Abstract

Background: Even if the management and treatment of patients with non-ST-elevation myocardial infarction (NSTEMI) have significantly evolved, it is still a burgeoning disease, an active volcano with very high rates of morbidity and mortality. Therefore, novel management and therapeutic strategies for this condition are urgently needed. Lately, theories related to the role of various blood cells in NSTEMI have emerged, with most of this research having so far been focused on correlating the ratios between various leukocyte types (neutrophil/lymphocyte ratio-NLR, neutrophil/monocyte ratio-NMR). But what about erythrocytes? Is there an interaction between these cells and leukocytes, and furthermore, can this relationship influence NSTEMI prognosis? Are they partners in crime? Methods: Through the present study, we sought, over a period of sixteen months, to evaluate the neutrophil/red blood cell ratio (NRR), monocyte/red blood cell ratio (MRR) and lymphocyte/red blood cell ratio (LRR), assessing their potential role as novel prognostic markers in patients with NSTEMI. Results: There was a statistically significant correlation between the NRR, LRR, MRR and the prognosis of NSTEMI patients. Conclusions: These new predictive markers could represent the start of future innovative therapies that may influence crosstalk pathways and have greater benefits in terms of cardiac repair and the secondary prevention of NSTEMI.

## 1. Introduction

An overview of the last few decades shows worrying numbers regarding cardiovascular mortality in patients with acute coronary syndrome (ACS) [1,2].

As of late, the management of these patients has significantly evolved. However, these steps forward have been accompanied by a step backward—with morbidity and mortality rates continuing to rise [3].

Research has previously explored the connection between the red blood cell distribution width, neutrophil-to-lymphocyte ratio, and other leukocyte ratios and the severity of myocardial infarction [4,5,6]. However, delving deeper into the immune response and inflammatory mechanisms during ACS is imperative.

The key processes in the pathophysiology of acute myocardial infarction are atherosclerotic plaque rupture and thrombosis, with both being promoted by inflammation [7,8].

Neutrophils, monocytes, lymphocytes and red blood cells are very important players in the onset and progression of inflammation during myocardial infarction [9,10].

The initial inflammatory phase is marked by the infiltration of polymorphonuclear neutrophils (PMNs), which is considered the most damaging stage following an AMI. In the subsequent phase, macrophages are recruited to the site, playing a protective role after the initial neutrophil response. These macrophages are responsible for engulfing debris and dead cells, aiding in the recovery of the heart muscle damaged by ischemia. Research has indicated that depleting macrophages post-AMI (acute myocardial infarction) can lead to increased mortality rates in animal studies. Regulatory T-cells are crucial in converting pro-inflammatory M1 macrophages into anti-inflammatory M2 macrophages, thereby influencing the immune response post-AMI and potentially improving healing outcomes in experimental scenarios [11,12,13].

Understanding the intricate interplay between various immune cell populations is crucial for deciphering the progression and consequences of myocardial infarction. Current blood ratio analyses have overlooked the role of erythrocytes in this context. 

By clarifying the connection between lymphocytes, neutrophils, monocytes, and erythrocytes in the context of MI, healthcare providers may improve their ability to intervene promptly and effectively, potentially leading to better patient outcomes and reduced morbidity and mortality rates.

The neutrophil/red blood cell ratio (NRR), monocyte/red blood cell ratio (MRR) and lymphocyte/red blood cell ratio (LRR) were first cited in 2019 and were correlated with the prognosis of advanced breast cancer [14]. Inspired by these findings, we sought to explore the prognostic potential of these biomarkers in ACS. To the best of our knowledge, until now, these ratios have not been investigated in relation to cardiovascular disease. 

The scientific and practical significance of these potential predictive markers is substantial. Furthermore, conducting larger randomized studies in the future may make it feasible to reduce the harm caused by ACS and ultimately improve the prognosis for these patients by addressing and reducing inflammation.

## 2. Materials and Methods

### 2.1. Study Design and Study Population

The present prospective, observational study was conducted in the Cardiology Department of the “Bagdasar-Arseni” Emergency Hospital in Bucharest, Romania, between January 2022 and April 2023. 

The study cohort constituted patients admitted with a diagnosis of NSTEMI and who subsequently underwent percutaneous coronary intervention (PCI). 

Non-ST elevation myocardial infarction was defined by symptoms consistent with acute coronary syndrome (ACS), electrocardiogram with myocardial ischemia signs—without changes consistent with STEMI, elevated cardiac biomarkers, and abnormal wall motion on cardiac ultrasonography. 

The inclusion criteria were patients over 18 years of age, of either sex, bearing the diagnosis of NSTEMI. The following exclusion criteria were used: (1) patients with hematological disease; (2) patients with severe liver or kidney disease; (3) patients with ongoing infection; (4) patients diagnosed with autoimmune diseases or malignancy; (5) patients having recently undergone chemotherapy/radiotherapy; and (6) patients with moderate and severe anemia.

A total of 41 individuals were removed from the study due to the exclusion criteria (12 had multiple organ failure, 16—pneumonia, 2—tuberculosis, 1—prior liver transplantation, 1—essential thrombocythemia, 1—frontal meningioma, 5—active SARS-CoV2 infection, 3—moderate or severe anemia).

In the end, a total number of 97 non-ST-elevation myocardial infarction patients were enrolled and evaluated.

Demographic data (age, sex, ethnicity) was recorded, along with clinical history, drug use and cardiovascular risk factors (arterial hypertension, diabetes mellitus, dyslipidemia, a BMI > 30 kg/m^2^, smoking). All the patients had venous blood samples collected within 10 min from admission. To perform the complete blood count, two milliliters of whole blood were collected in a vacutainer tube containing 3.6 mg of EDTA (ethylene-diamine-tetra-acetic acid). Subsequently, the blood underwent analysis using a Beckman Coulter AcT 5diff AL autoanalyzer to determine the hematological parameters. The NT-proBNP level was evaluated through the Elisa method with a PATHFAST compact immunoanalyzer autoanalyzer. 

The laboratory tests included a complete blood count, biomarkers of cardiac injury, admission high-sensitivity troponin I and peak high-sensitivity troponin I, admission creatine kinase (CK), creatine kinase-MB (CK-MB) and peak CK, CK-MB-serum electrolytes, serum creatinine and glucose, ESR, fibrinogen, and NT-proBNP. Renal dysfunction was defined as an elevated serum creatinine >1.3 mg/dL for men and >1.1 mg/dL for women. Regarding NT-proBNP, which is a marker of cardiac decompensation, an elevated level was defined as values >300 pg/mL.

The NRR was defined by the absolute neutrophil count divided by the absolute red blood cell count, the MRR was calculated by dividing the absolute count of monocytes by the absolute count of red blood cells, and the LRR was obtained by dividing the absolute count of lymphocytes by the absolute count of red blood cells. The LVEF (left ventricular ejection fraction) values were obtained using transthoracic echocardiography performed during the index hospitalization. The modified Simpson biplane method was used to calculate the left ventricular end-diastolic volumes (LVEDVs) and end-systolic volumes (LVESVs) from 4- and 2-chamber views. The LV volumes were corrected for body surface areas.

All the demographic, clinical and laboratory data that we have attempted to correlate with the evolution of NSTEMI patients are presented in Table 1 and Figure 1. 

The endpoints of our study were established as the occurrence of Killip class III/IV, LVEF, indication for surgical revascularization, in-hospital mortality, 30-day mortality, and 30-day readmission. Concerning the definitions, Killip class III refers to those patients with evident acute pulmonary edema, while Killip class IV defines patients with cardiogenic shock (hypotension—a systolic blood pressure under 90 mmHg and signs of low cardiac output) [15]. The decision regarding surgical revascularization was mainly guided by the specific characteristics of the coronary artery lesion. Factors such as the location, size and complexity of the lesion significantly influenced the decision regarding the need for surgery. It is important to emphasize that this decision was personalized and involved the input of a multidisciplinary team of healthcare professionals, including cardiologists and cardiac surgeons. In our investigation, the primary method used for the functional assessment of coronary artery stenosis was the quantitative flow ratio (QFR).

Fractional flow reserve (FFR) measurements were carried out for intermediate-grade lesions. For all the coronary arteries, at least 2 plain images were taken. Also, at least two cardiologists regularly reviewed the coronary angiograms. The periprocedural management and pharmacotherapy were guided by the latest ESC guidelines on myocardial infarction [15].

The follow-up consisted of quantifying the 30-day readmission, survival, and 30-day all-cause mortality. This was made possible by consulting the online portal of the Romanian national health authority.

This study was conducted according to the guidelines of the Declaration of Helsinki and approved by the Ethics Committee of “Carol Davila” University of Medicine and Pharmacy, Bucharest, Romania (protocol code PO-35-F-03, date 1 October 2021).

### 2.2. Statistical Analysis

The statistical findings were obtained by using logistic and linear regression models. The statistical results were visually represented using boxplots, distribution representations and receiver operating characteristic (ROC) curves. To determine the statistical significance of the regression model, a *p*-value threshold of 0.05 was used. If the *p*-value was below this threshold, the model was considered statistically significant. Based on the analysis of the regression function, a mathematical model, similar with an equation, was developed. This formula included a score for the event happening, estimated coefficients for the NRR, LRR, and MRR and the corresponding NRR, LRR, and MRR values. To convert the score value into a probability, the following equation was applied: Probability = $\frac{1}{1+e^{-score}}$.

To analyze the cut-off values of the NRR, LRR and MRR in predicting an event, the receiver operating characteristic (ROC) curve was used.

## 3. Results

Our research involved 97 individuals diagnosed with NSTEMI. All the patients were of Caucasian ethnicity. There was a higher proportion of male patients (63.92%) than female patients (36.08%). The average age was found to be 64 years old. We assessed common cardiovascular risk factors, concluding that 39.18% of individuals were diabetic, 56.7% had dyslipidemia, 36.08% had a BMI over 30 kg/m^2^ and 79.38% had arterial hypertension. Furthermore, 18.56% of patients were classified as being Killip class III/IV (Table 1).

In terms of the LRR, throughout most of our cohort, the values were found to be below 0.001. Only a few patients were shown to have higher values, including one with an LLR of around 0.003. Most patients had an NRR between 0.001 and 0.003, and an MRR between 0.0001 and 0.0003 (Figure 1). 

### 3.1. NRR and Its Predictive Value in NSTEMI Prognosis

#### 3.1.1. Correlation between NRR and Laboratory Parameters

In our research, we did not find any statistically significant correlation between the NRR and the admission levels of CK-MB, admission high-sensitivity troponin I as well as the peak of CK, CK-MB or creatinine levels, as presented in Table 2. However, we discovered a statistically significant correlation between the NRR level and increased left ventricular stretch, as measured by NT-proBNP, as well as myocardial damage evidenced by peak high-sensitivity troponin I (Table 2).

Figure 2 and Figure 3, respectively, depict the visual representation of the linear regression analysis performed on the NRR and NT-proBNP and the NRR and peak high-sensitivity troponin I, providing further insight into their relationship. 

#### 3.1.2. Correlation between NRR and Clinical Outcomes

Regarding the association of the NRR with clinical outcomes, we did not demonstrate any significant correlation between the NRR and triple vessel disease, length of hospital stays, in-hospital mortality, 30 day-mortality or 30-day readmission. However, we found a statistically significant correlation between the NRR level and the risk of developing Killip class III/IV, as well as the necessity of surgical revascularization and LVEF (Table 3).

Figure 4 shows the linear relationship between the NRR and LVEF.

As indicated in Table 4, a statistical correlation between the higher NRR value and Killip class III/IV is supported by the following. (1) The NRR *p*-value was 0.055. (2) The R square value of 0.039 shows that the variation in the NRR values accounts for approximately 3.9% of the variability in Killip class III/IV developing. The NRR has a small impact on the likelihood of Killip class III/IV occurrence, but the *p* value is very close to our threshold of 0.05, which would suggest that a significant relationship exists. However, further studies with larger sample sizes are needed to confirm these findings. 

By fitting a logistic regression, we came up with a model. The formula used to calculate the probability score for an NSTEMI patient developing Killip class III/IV based on the NRR value is: −2.6494 + 658.36 × NRR value. For example, if we have a patient with an NRR value of 0.0004, the score would be −2.38. By utilizing the logistic function, the probability derived from a score of −2.38 is 0.084, or more plainly put, there is a 8.4% chance that an NSTEMI patient with an NRR value of 0.0004 will be classified as Killip class III/IV. Boxplots were constructed to visualize the distribution of the NRR in both groups: NSTEMI with Killip class III/IV and NSTEMI without Killip class III/IV (Figure 5a).

The boxplot for the Killip class III/IV group revealed a lower whisker of 0.0004, a lower quartile of 0.0013, a median of 0.0020, an upper quartile of 0.0026 and an upper whisker of 0.0037. In contrast, the boxplot for patients without Killip Class III/IV showed a lower whisker of 0.0004, a lower quartile of 0.0012, a median of 0.0014, an upper quartile of 0.0019, and an upper whisker of 0.0029. Thus, the boxplot analysis clearly demonstrates a notable difference in the NRR distribution between NSTEMI patients with Killip class III/IV and those with a lower Killip class (Figure 5a). The NRR values in the Killip class III/IV group exhibit a wider range and are higher overall compared to the non-Killip class III/IV group. These differences in the NRR between NSTEMI patients with Killip class III/IV and those without suggest that the NRR may serve as a potential indicator of disease severity.

Moreover, it is evident from Figure 5b that patients with NSTEMI and a greater NRR value tend to develop Killip III/IV (the red area, representing Killip class III–IV patients, extends beyond the blue area—patients who are not categorized as being Killip class III/IV). 

Figure 6 illustrates the estimated probability of Killip III/IV occurring for the NRR values.

According to the logistic regression model, as the NRR value increases, so does the likelihood that a patient’s evolution will be consistent with Killip class III/IV. For instance, patients with an LRR value of 0.0035 had a 40% chance of ultimately being categorized as Killip class III/IV (Figure 6).

The optimal cut-off value of the NRR in terms of Killip class III/IV was determined by employing the Youden index from the ROC results. An NRR of 0.00436 was found to be the best threshold, with optimal sensitivity and specificity (Figure 7).

Regarding the association between the NRR and the necessity of surgical revascularization, we initially identified a statistically significant positive correlation (Table 3). To delve deeper into this relationship, we further examined the cut-off value for the NRR in predicting the necessity of surgical revascularization.

Using the Youden index from the ROC results, we identified an NRR threshold of 0.0036 as the most effective, providing optimal sensitivity and specificity (Figure 8).

### 3.2. LRR and Its Predictive Value in NSTEMI Prognosis

#### 3.2.1. Correlation between LRR and Laboratory Parameters

Within our study, we did not find a statistically significant association between the LRR and the extent of the myocardial infarction (measured by the admission CK, CK-MB, high-sensitivity troponin I or peak CK, CK-MB, peak high-sensitivity troponin I). Likewise, there was no correlation observed between the LRR and left ventricular stretch (measured by NT-proBNP), or the LRR and kidney dysfunction, in contrast to the previous associations we demonstrated between the NRR and these parameters (Table 5).

#### 3.2.2. Correlation between NRR and Clinical Outcomes

For the clinical outcomes, our study results revealed a statistically significant correlation between a high LRR and in-hospital mortality, as well as a higher risk of developing Killip class III/IV. However, no statistically significant relationship between the LRR and other outcomes was found (Table 6). 

According to the table provided below (Table 7), there is a strong link between a higher LRR value and Killip class III/IV.

This correlation is supported by the following information. 1. The *p* value for the LRR is 0.043, indicating statistical significance. 2. The R square value of 0.049 suggests that approximately 4.9% of the variability in developing Killip class III/IV can be attributed to variations in the LRR values. The formula used to calculate the probability score for an NSTEMI patient developing Killip class III/IV based on the LRR value is: −2.2424 + 1203.1063 × LRR value. For instance, if a patient had an LRR value of 0.0007, the score would be −1.400. By applying the logistic function, the probability derived from a score of −1.400 is 0.197, or in other words, there is a 19.7% chance that an NSTEMI patient with an LRR value of 0.0007 will be categorized as Killip class III/IV.

The boxplot in Figure 9a provides a visual representation of the distribution of the LRR ratios in both groups—those with Killip class III/IV and those found outside of that category.

It is evident that patients with Killip class III/IV have a wider range of lymphocyte-to-erythrocyte ratio compared to those without Killip class III/IV. For patients with Killip III/IV, the lower whisker of the boxplot is at 0.0003, indicating that the lowest observed value for the lymphocyte-to-erythrocyte ratio in this group is 0.0003. The lower quartile, which represents the 25th percentile of the data, is at 0.0005. This means that 25% of the patients with Killip class III/IV have a lymphocyte-to-erythrocyte ratio of 0.0005 or lower. The upper quartile, representing the 75th percentile, is at 0.0010. This indicates that 75% of the patients with Killip class III/IV have a lymphocyte-to-erythrocyte ratio of 0.0010 or lower. The upper whisker, which represents the maximum value within 1.5 times the interquartile range, is at 0.018. This suggests that there are some extreme values in the data, with a few patients having a much higher lymphocyte-to-erythrocyte ratio. Overall, the boxplot reveals that patients with Killip class III/IV have a wider range of lymphocyte-to-erythrocyte ratio compared to those without Killip class III/IV. This suggests that the presence of Killip class III/IV may be associated with higher values of this ratio (Figure 9a).

We used Figure 9b to illustrate how plotting the density for the LRR grouped by Killip class III/IV can be a useful way of observing the distribution of NSTEMI patients with Killip class III/IV versus NSTEMI patients without Killip class III/IV based on the LRR value. It is evident from Figure 9b that patients with NSTEMI and a greater LRR value tend to develop Killip III/IV (the red area, representing Killip class III/IV patients, extends beyond the blue area—patients who are not categorized as being Killip class III/IV). 

In the ROC analysis (Figure 10), the optimal cut-off value for the LLR is 0.0023 and this is associated with a threshold of probability of 0.55. This LLR value maximizes the difference between the True Positive Rate and False Positive Rate. So, this value of 0.0023 could be used to determine if a patient is more likely to have a severe Killip class (III or IV) or not. If an observed value of the LLR is above 0.0023, it is more likely for him to develop Killip III or IV.

In our study, we found that a high LRR was statistically correlated with the in-hospital mortality. Building upon this finding, we proceeded to determine the optimal cut-off value for the LRR by employing the Youden index from the ROC results. Our analysis revealed that an LRR of 0.0047 served as the best threshold, offering optimal sensitivity and specificity (Figure 11).

### 3.3. MRR and Its Predictive Value in NSTEMI Prognosis

#### Correlation between MRR and Laboratory Parameters

In our study, we identified a statistically significant positive correlation between the MRR and left ventricular stretch (measured by NT-proBNP), as well as the MRR and kidney dysfunction. However, we did not observe any statistically significant association between the MRR and myocardial damage (quantified by the admission and peak high-sensitivity troponin I levels as well as admission and peak myocardial necrosis enzymes) (Table 8).

Figure 12 and Figure 13, respectively, show the linear regression relationship between the MRR and NT-proBNP and between the MRR and creatinine. 

Regarding the clinical outcomes, we observed a statistically significant correlation between the MRR and the risk of developing Killip class III/IV, as well as the necessity of surgical revascularization.

However, we did not find any statistically significant association between the MRR and LVEF, triple vessel disease, length of hospitalization, in-hospital mortality, 30-day mortality or 30-day readmission (Table 9).

A visual representation of the distribution of the MRR within two groups (NSTEMI patients with Killip class III/IV and NSTEMI patients without Killip class III/IV) was created using boxplots (Figure 14a). When comparing the two groups of patients, the patients with Killip class III/IV showed significantly higher absolute MRR values. These findings suggest that the MRR may serve as a potential indicator of disease severity in NSTEMI patients (Figure 14b).

Based on the table provided below (Table 10), there is a significant relationship between higher MRR values and Killip class III/IV. 

This correlation is supported by the statistical significance of the *p*-value for the MRR (0.033) and the R-square value (0.049), indicating that approximately 4.9% of the variability in the development of Killip class III/IV can be attributed to differences in the MRR values. 

The formula used to calculate the probability score for an NSTEMI patient developing Killip class III/IV based on the MRR value is: −2.8215 + 7508.73253 × MRR value.

For example, if a patient had an MRR value of 0.0006, the score would be 1.68. 

The probability derived from a score of 1.68, when applying the logistic function ($\frac{1}{1+e^{-score}}$), is approximately 0.843, or 84.3%. This means that there is an 84.3% chance that an NSTEMI patient with a score of 1.68 will be in Killip class III/IV.

Finally, we have used the ROC curves to establish the optimal cut-off—that value of MRR above which there is a high probability of NSTEMI patients developing Killip class III/IV. As shown in Figure 15, the cut-off value of the MRR was found to be 0.000402.

Furthermore, after demonstrating the statistically positive correlation between the MRR and the necessity of surgical revascularization, we have used the ROC curves to establish the optimal cut-off for the MRR—that value of MRR above which there is a high probability of NSTEMI patients needing surgical revascularization. As shown in Figure 16, the cut-off value for the MRR was found to be 0.000329.

## 4. Discussion

To the best of our knowledge, this is the first study that attempted to find and found a correlation between the ratios of monocytes, lymphocytes and neutrophils to erythrocytes and the prognosis of acute coronary syndrome.

### 4.1. Neutrophils—The First Pillar of a Bottom-Up Phenomenon: Insight into the Role of Neutrophils in Acute Coronary Syndrome and Their Interaction with Red Blood Cells 

Our study revealed a statistically significant positive association between the neutrophil-to-red blood cell ratio (NRR) and myocardial damage, as indicated by the peak high-sensitivity troponin I levels. Despite the lack of a statistical correlation between the NRR and CK or CK-MB, we can conclude that the NRR is linked to myocardial injury, given the superior sensitivity of high-sensitivity troponin I compared to myocardial necrosis enzymes.

Additionally, previous research has suggested a connection between the size of a myocardial infarction and the neutrophil count, which further reinforces the findings of our study. One study by Dogan et al. highlighted the positive correlation between an elevated neutrophil count and increased myocardial damage, as evident from the elevated levels of CK-MB, cardiac troponin, and scintigraphic infarct size [16].

We also observed a statistically positive correlation between the NRR and increased left ventricular stretch, as measured by the NT-proBNP levels. Furthermore, we found that higher NRR values were associated with a higher risk of acute heart failure (Killip class III/IV), severe coronary lesions requiring surgical revascularization, and a lower left ventricular ejection fraction (LVEF). These findings are consistent with previous studies suggesting that elevated neutrophil counts are linked to worse prognostic outcomes in acute myocardial infarction [17,18].

NT-proBNP, a cardiac neurohormone, is released in response to increased stretching of the left ventricular wall and is further stimulated by myocardial ischemia [19].

Our results further support a statistically significant positive correlation between the NT-proBNP and neutrophil count (expressed by NRR), as previously reported in the literature [20,21].

Several previous studies have also demonstrated a statistically significant association between the neutrophil count and the severity of coronary artery disease, particularly in predicting the presence of left main/three vessel disease in patients with non-ST-elevation myocardial infarction [22,23]. Our study reinforces these findings and adds to the growing body of evidence supporting the role of the NRR as a potential biomarker for cardiovascular risk assessment.

When studying the correlation between the neutrophil-to-red blood cells ratio (NRR) and myocardial contraction, we demonstrated a statistically inverse correlation with the LVEF. We found support in the existing literature for the negative inotropic effects on cardiac tissue caused by a high pro-inflammatory state, as indicated by an increased number of neutrophils. This pro-inflammatory status can disrupt cardiac metabolism, contribute to myocardial remodeling, and ultimately, lead to heart failure [24].

While our study did not identify a correlation between the NRR and in-hospital mortality or 30-day mortality, it is important to note that the limited sample size of our cohort may have hindered the detection of these correlations. Larger studies with a greater number of participants are needed to provide more conclusive insights into this aspect. 

Furthermore, if we delve deeper into the pathophysiological mechanisms, we can uncover intriguing relationships, not only regarding neutrophils but also, more significantly, the association between erythrocytes and neutrophils during acute coronary syndrome. Exploring these relationships may shed further light on the complex interplay between different cellular components in cardiovascular disease progression.

Neutrophils are the first leukocytes to enter the site of the myocardium affected by ischemia [25,26].

Initially, neutrophils recruited to the area of myocardial infarction phagocytose and remove cellular debris. Simultaneously, they release myeloperoxidase and protease, which exacerbate vascular and tissue injury [27,28]. Additionally, neutrophils produce oxygen free radicals via an NADPH-dependent mechanism (nicotinamide adenine dinucleotide phosphate) [29].

This mechanism may result in direct myocardial injury through alterations to lipids, proteins and amino acids [30]. Recent studies have highlighted the ability of neutrophils to form NETs (extracellular traps). Likewise, NETs can activate the NLRP3 inflammasome in macrophages, leading to the release of IL-1 417 beta and IL-18 [31,32]. Originally associated with sepsis and neoplasia, NETs have also been found to play a role in cardiovascular diseases, and their presence was actually detected within the culprit lesions of acute coronary syndrome [33]. As to the ways neutrophils and red blood cells affect one another, there are multiple studies that showcase the interactions between NETs and red blood cells, interactions which can result in erythrocyte deformation and even fragmentation [34]. Our study reinforced the previous theories according to which a higher number of neutrophils is associated with a worse prognosis, and furthermore, proved that there is another noteworthy explanation for this phenomenon apart from the data already known: the interaction between erythrocytes and neutrophils. 

### 4.2. Lymphocytes—The Second Pillar of the Bottom-Up Phenomenon 

Our study demonstrated a statistically significant correlation between a high lymphocyte-to-red blood cell ratio (LRR) value and an increased risk of developing Killip class III/IV. Previous studies support our results, demonstrating that a high lymphocyte count is correlated with an unfavorable prognosis [35].

However, contrasting results from other studies suggest that a low lymphocyte count is associated with a worse prognosis in acute coronary syndrome. The underlying mechanisms for these disparate outcomes are not fully understood. One theory that may explain these contradictory results is the variable pro- or anti-atherogenic and inflammatory properties of T and B cell subsets [36,37]. Considering the complexity of the immune response, the prognostic importance of these specific subtypes may outweigh the significance of the overall lymphocyte count. Future studies utilizing advanced multicolor flow cytometry techniques should prioritize investigating the role of these subtypes as potential biomarkers. There are studies revealing a connection between an elevation in pro-atherogenic CD4 CD28hull T cells and a higher incidence of recurrent cardiovascular events [38,39,40].

This finding may explain our results, reinforcing the hypothesis that the activation of these specific T cell subsets contributes to an unfavorable prognosis in CAD. Furthermore, our study demonstrates a statistically significant and direct relationship between a high LRR level and in-hospital mortality, indicating the prognostic value of the LRR in predicting patient outcomes. However, no statistically significant association was found between the LRR and NT-proBNP level, left ventricular ejection fraction (LVEF), or creatinine in our study population. Delving deeper into the pathophysiological mechanisms of acute coronary syndrome may reveal intriguing relationships, not only involving lymphocytes but also the association between erythrocytes and lymphocytes. Understanding these interplays can provide valuable insights into the complex immune response and its impact on myocardial injury.

While neutrophils are part of the innate immune response and are responsible for initiating the inflammatory cascade, lymphocytes make up the second wave of immune cells to arrive at the site of myocardial injury, being part of the adaptive immune response and playing an important regulatory role [41,42].

Tissue antigens from the affected area, together with the released pro-inflammatory cytokines, cause the activation of B lymphocytes, which will produce Ccl17 through Myd88 and/or the Trif signaling pathway. In this way, monocytes are recruited from the periphery to the area of acute myocardial infarction [43]. Experimental studies have indicated that a decrease in the production of Ccl17 and, implicitly, a decrease in the number of recruited monocytes can be achieved through the blocking of B lymphocytes using monoclonal antibodies attached to CD20. In terms of the LRR ratio, in the last 30 years, extensive research has revealed the modulatory role of red blood cells in the proliferation of T lymphocytes, both in vitro and in vivo [44].

Studies have demonstrated that, in healthy individuals, red blood cells prevent the full maturation of dendritic cells, so that dendritic cells remain at an immature/tolerogenic stage, thus forestalling an inflammatory process. When red blood cells enter an area of ischemia and high oxidative stress, their structure undergoes a series of changes (rearrangement of the cytoskeleton, loss of asymmetry and lipid layers) [45]. RBCs that have suffered from oxidative damage (like in myocardial infarction) can increase the proliferation of mitogen-driven T cells, as well as apoptosis and a Th1 proinflammatory cytokine response (due to these RBCs’ inability to prevent the maturation of dendritic cells) [46,47].

The theories presented above are supported by our study, in which an increased lymphocyte-to-red blood cell ratio was correlated with a poor prognosis in NSTEMI patients.

### 4.3. Monocytes—The Third Pillar of the Bottom-Up Phenomenon 

Our study revealed that the monocyte-to-red blood cell ratio was associated with increased left ventricular stretch, as measured by NT-proBNP levels, severe post myocardial infarction heart failure (Killip class III/IV) and kidney dysfunction.

Firstly, the correlations between a high MRR level and increased left ventricular stretch and severe post-myocardial infarction heart failure, respectively, are supported by studies highlighting the presence of different subsets of monocytes, with each subset potentially contributing positively or negatively to heart failure (HF) pathogenesis. Monocytosis, characterized by an increased number of monocytes, has been associated with left ventricular dysfunction following myocardial infarction. This is attributed to the release of various cytokines, such as IL-1α, IL-1β, IL-6, and TNF-α, which negatively affect myocardial healing and contribute to the development of heart failure. Despite this understanding, the precise mechanisms and intricate details of these processes have not yet been fully elucidated [48]. It is important to emphasize that the role of monocytes in cardiovascular diseases is multifaceted and extends beyond inflammation. Monocytes also play a role in processes related to regeneration, repair, and modulation of the prothrombotic state [9,49].

Secondly, the statistically direct correlation between a high monocyte-to-red blood cell ratio (MRR) and the need for surgical revascularization aligns with previous studies. Accumulation of monocytes and monocyte-derived phagocytes within the arterial wall has been observed in prior studies, and they actively contribute to the chronic inflammatory process and the development, progression, and complications of atherosclerosis. Monocytes have the ability to migrate to the artery wall, differentiate into macrophages, and initiate the secretion of matrix metalloproteinases, proinflammatory cytokines, and reactive oxidative species [50,51,52]. This finding suggests that an elevated MRR may serve as a potential indicator of the severity of coronary artery disease and the necessity of invasive procedures. 

Furthermore, our study identified a significant correlation between a high MRR and kidney dysfunction. The pathophysiology of kidney dysfunction is believed to be multifactorial, and several studies have suggested that the phenomenon of ischemia–reperfusion, along with inflammation, plays important roles in its development [53]. Recent discoveries have also shed light on significant phenotypic variations among human monocytes, indicating their potential relevance to the development of kidney dysfunction. Monocytes express distinct cytokines and adhesion molecules that play a crucial role in the inflammatory processes associated with the pathogenesis of kidney dysfunction [54,55].

It is worth noting that while our study yielded positive and significant results for the correlations mentioned above, we did not find a statistically significant relationship between a high MRR and in-hospital mortality or 30-day mortality. 

To achieve a better understanding of the correlation presented above, it is crucial to delve deeper into the pathophysiological mechanisms of acute coronary syndrome. By doing so, we can uncover intriguing relationships that not only involve monocytes but also demonstrate a significant association between erythrocytes and monocytes. Understanding these interplays can provide valuable insights into the complex immune response and its impact on myocardial injury.

We hypothesized above that, through Myd88 and/or Trif signaling, Ccl17-producing B lymphocytes stimulate the recruitment of monocytes [56,57].

Regarding the MRR, once red blood cells are exposed to oxidative stress (for example, in ACS), they polarize macrophages toward a proinflammatory-M1 phenotype, inducing the secretion of proinflammatory cytokines [41]. This theory is strengthened by our research, in which the higher the monocyte-to-red blood cell ratio, the more frequent the complications in NSTEMI patients.

Our comprehensive study showed that the presence of an increased NRR, LRR and MRR is associated with a poor prognosis in non-ST-elevation myocardial infarction, partially due to the high neutrophil, monocyte, and lymphocyte count but also owing to these leukocytes’ interaction with red blood cells. Through a bottom-up analytical framework, a complex process can be observed, in which all the pillars are connected to each other. As for the observation of severe Killip in our study, although this classification offers valuable clinical information, it may not provide a complete understanding of the underlying pathophysiology and prognosis. By identifying biological markers such as the NRR, MRR, and LRR, we can gain further insights into the disease mechanisms. Moreover, these data have the potential to enhance risk stratification, inform personalized treatment decisions, and guide the development of targeted therapies.

To the best of our knowledge, this is the first study that attempts to find a correlation between the ratios of monocytes, lymphocytes and neutrophils to erythrocytes and the prognosis of acute coronary syndrome. By analyzing the existing literature, we concluded that these links are based on interesting interactions between different blood cells. 

### 4.4. Limitations of the Study 

As for the limitations of this study, two major ones were its single-center design and its small patient cohort, highlighting the necessity for further well-structured studies to validate our findings. Additionally, the lack of long-term prognosis monitoring in these patients could impede the predictive significance of these biomarkers in the long run. Furthermore, our study population demographics revealed solely Caucasian origins, like most major ACS clinical score trials.

Moreover, since the precise moment in the evolution of acute coronary syndrome when the peak inflammatory response occurs is still unknown, choosing the best temporal approach to blood sample acquisition proved challenging. Despite all these limitations, our study serves as an initial step toward subsequent studies and lays the groundwork for innovation in ACS management.

#### Future Directions

Further medication-related breakthroughs, as well as more effective anti-inflammatory treatments, may result from the insights gained from these investigations. There is a need for larger randomized studies targeting acute coronary syndrome; this is due to the abundance of preclinical and observational findings pertaining to the negative effects that inflammation has on ACS patients’ prognosis. Future research ought to determine whether reducing inflammation can improve the outcomes in NSTEMI patients.

## 5. Conclusions

Our study findings highlight the pivotal role of the NRR, MRR, and LRR in predicting patient outcomes following NSTEMI, paving the way for the discovery of novel biomarkers to assess risk and forecast prognosis. By delving into these markers, tailored treatment plans can be crafted for individuals grappling with MI. Understanding the intricate interplay among lymphocytes, neutrophils, monocytes, and erythrocytes in the context of MI holds the potential to enhance healthcare providers’ ability to intervene swiftly and effectively, thereby enhancing patient outcomes and curtailing morbidity and mortality rates.

Moreover, due to their simplicity, affordability, accessibility, and cost-effectiveness, the NRR, MRR and LRR can be effectively used to anticipate complications in NSTEMI patients, helping to determine the most suitable course of management. 

Further exploration of the efficacy of anti-inflammatory drugs in NSTEMI patients presents a promising avenue for future research, as dampening inflammation could potentially lessen the impact of acute coronary syndrome. 

## Figures and Tables

**Figure 1 healthcare-12-00824-f001:**
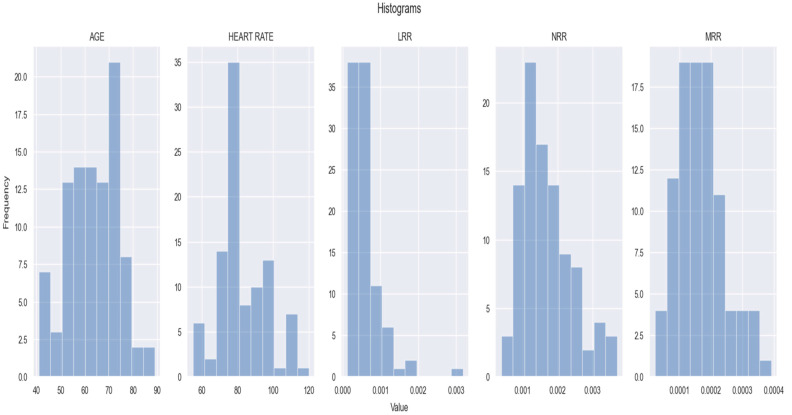
Clinical and main laboratory findings for the study population. From left to right, the distributions of the age, heart rate, LRR value, NRR value and MRR value among NSTEMI individuals. The figure is an original contribution by the authors.

**Figure 2 healthcare-12-00824-f002:**
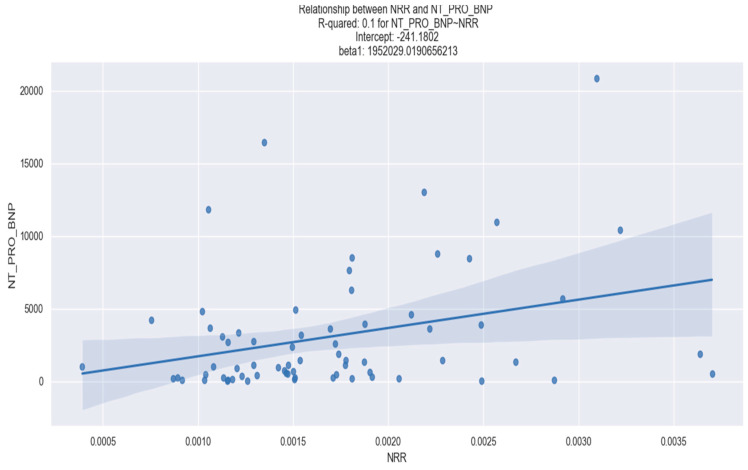
Linear regression between NRR and NT-proBNP. The figure is an original contribution by the authors.

**Figure 3 healthcare-12-00824-f003:**
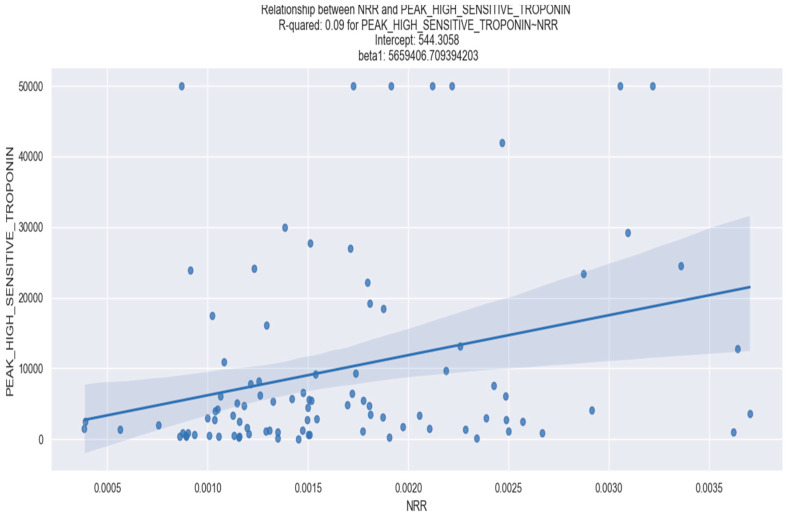
Linear regression between NRR and peak high-sensitivity troponin I. The figure is an original contribution by the authors.

**Figure 4 healthcare-12-00824-f004:**
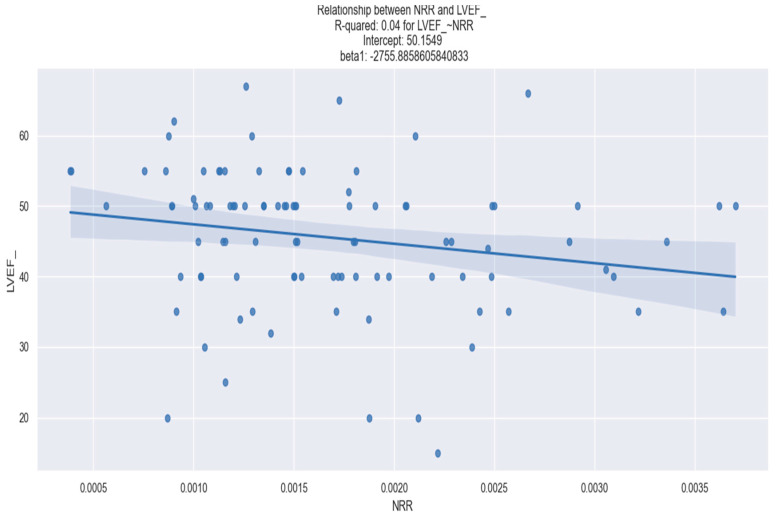
Linear regression between NRR and LVEF. The figure is an original contribution by the authors.

**Figure 5 healthcare-12-00824-f005:**
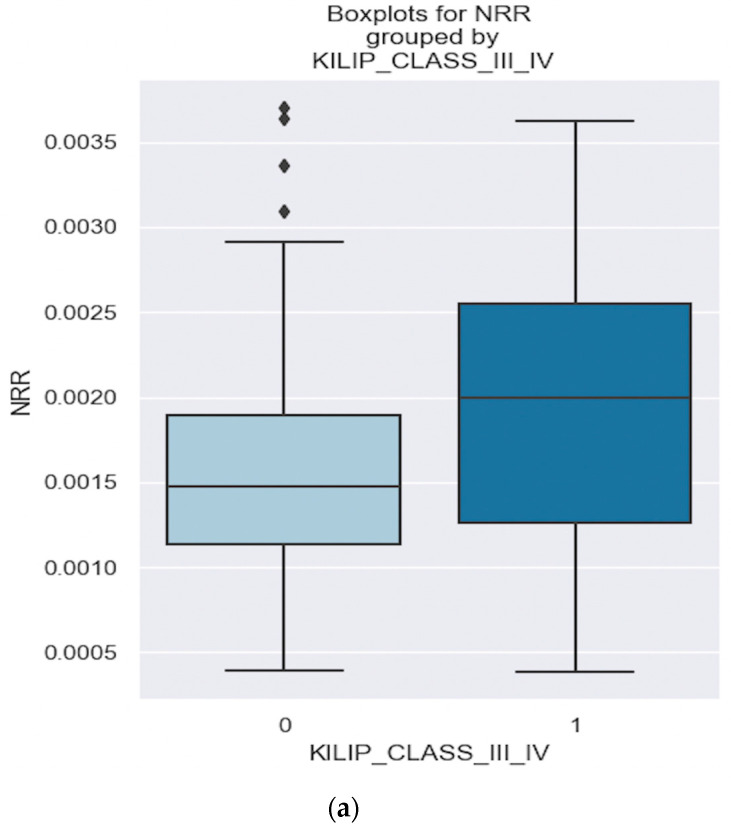
(**a**,**b**). Comparative view of NRR levels in patients with and without Killip class III/IV. The figure is an original contribution by the authors.

**Figure 6 healthcare-12-00824-f006:**
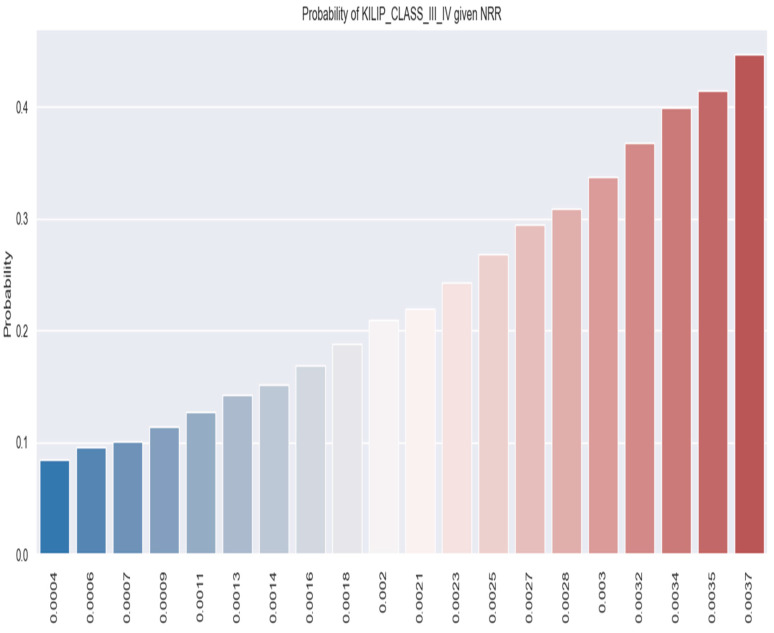
Probability of developing Killip class III/IV based on NRR values by applying the logistic regression model. The figure is an original contribution by the authors.

**Figure 7 healthcare-12-00824-f007:**
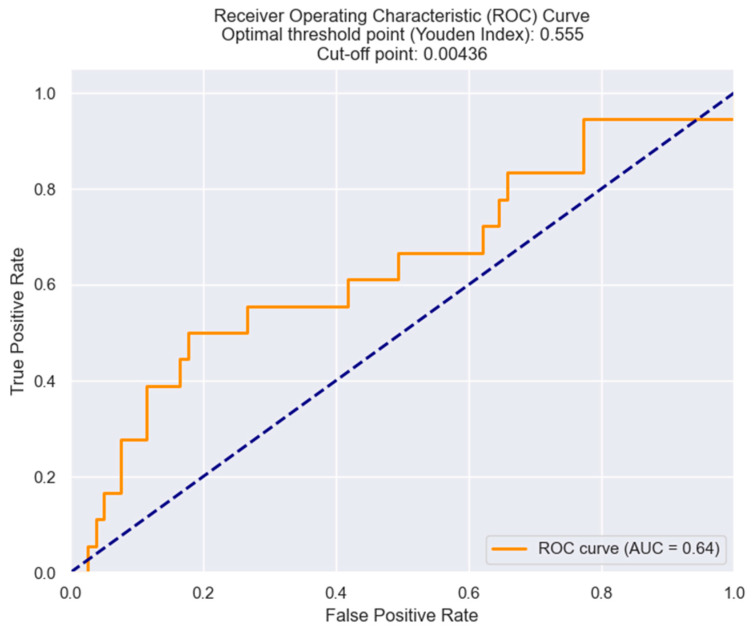
Receiver operating characteristic analysis and curve for predicting Killip class III/IV based on the NRR value. The figure is an original contribution by the authors.

**Figure 8 healthcare-12-00824-f008:**
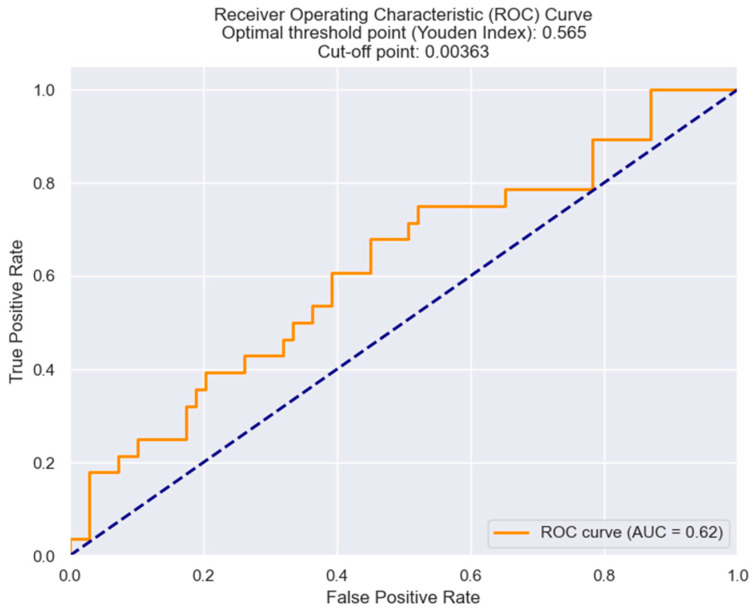
Receiver operating characteristic analysis and curve for predicting the necessity of surgical revascularization based on the NRR value. The figure is an original contribution by the authors.

**Figure 9 healthcare-12-00824-f009:**
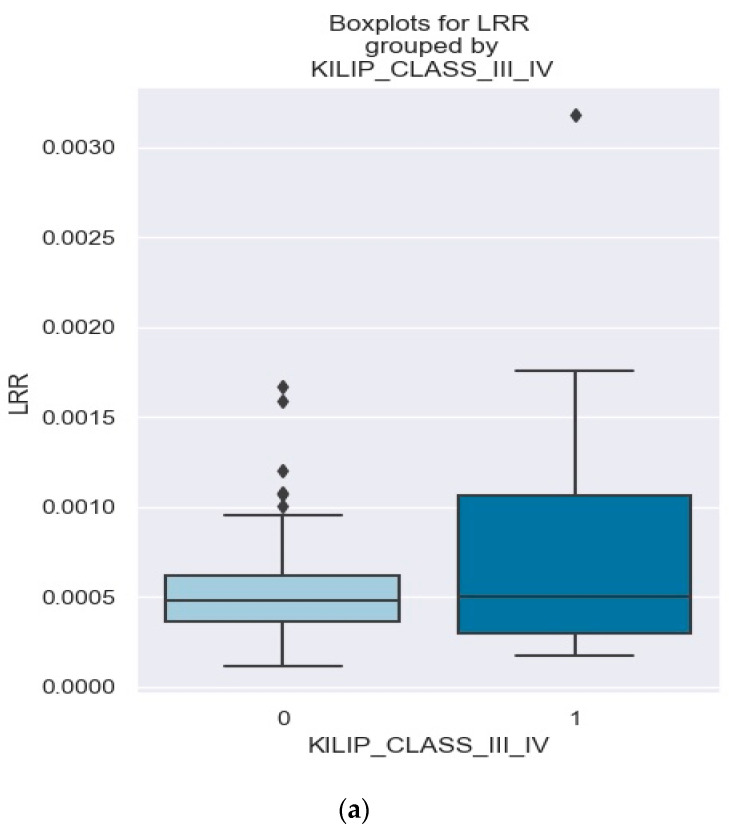
(**a**,**b**). Comparison of LRR levels between patients with and without Killip class III/IV. The figure is an original contribution by the authors.

**Figure 10 healthcare-12-00824-f010:**
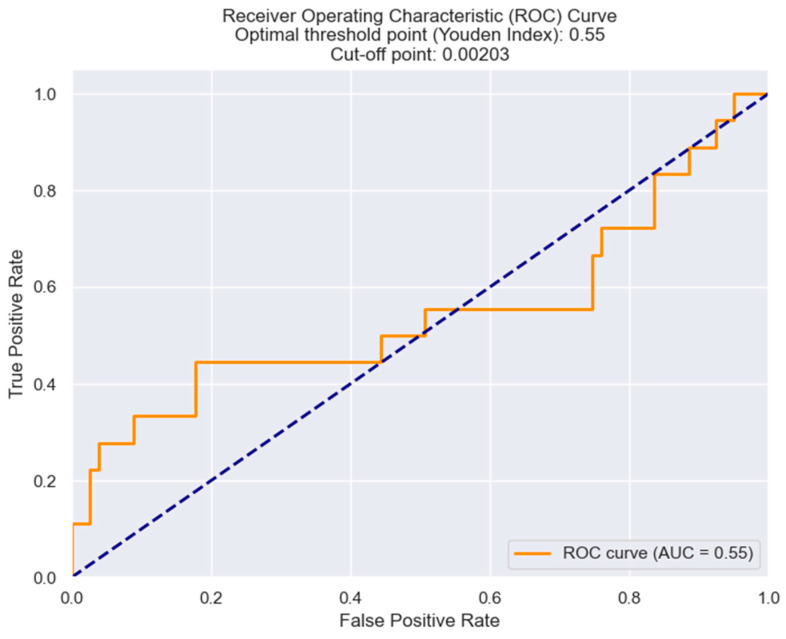
Receiver operating characteristic analysis and curve for predicting Killip class III/IV based on the LRR value. The figure is an original contribution by the authors.

**Figure 11 healthcare-12-00824-f011:**
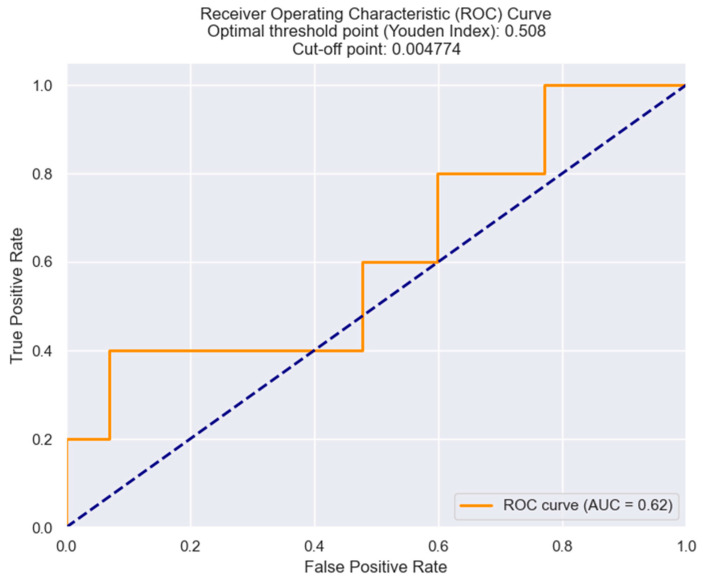
Receiver operating characteristic analysis and curve for predicting in-hospital mortality based on the LRR value. The figure is an original contribution by the authors.

**Figure 12 healthcare-12-00824-f012:**
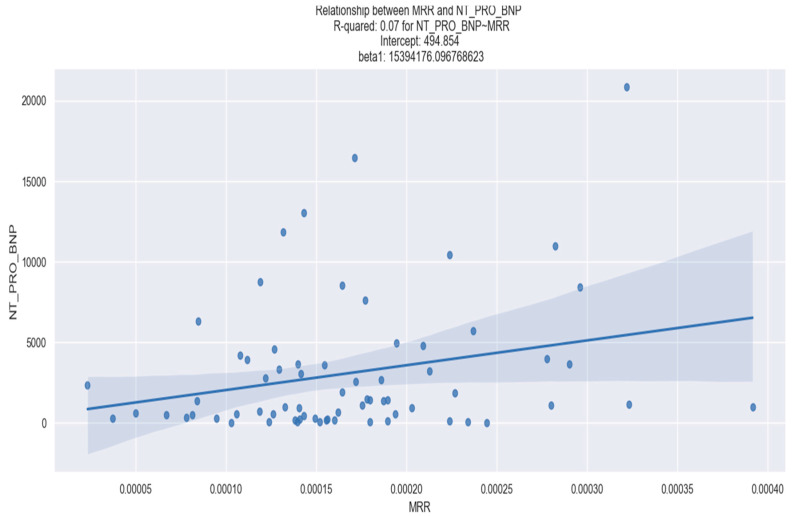
Linear regression between MRR and NT-proBNP. The figure is an original contribution by the authors.

**Figure 13 healthcare-12-00824-f013:**
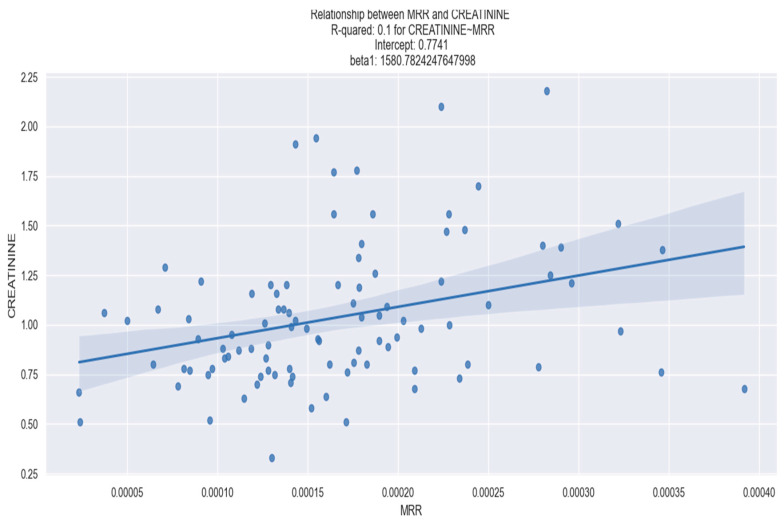
Linear regression between MRR and creatinine. The figure is an original contribution by the authors.

**Figure 14 healthcare-12-00824-f014:**
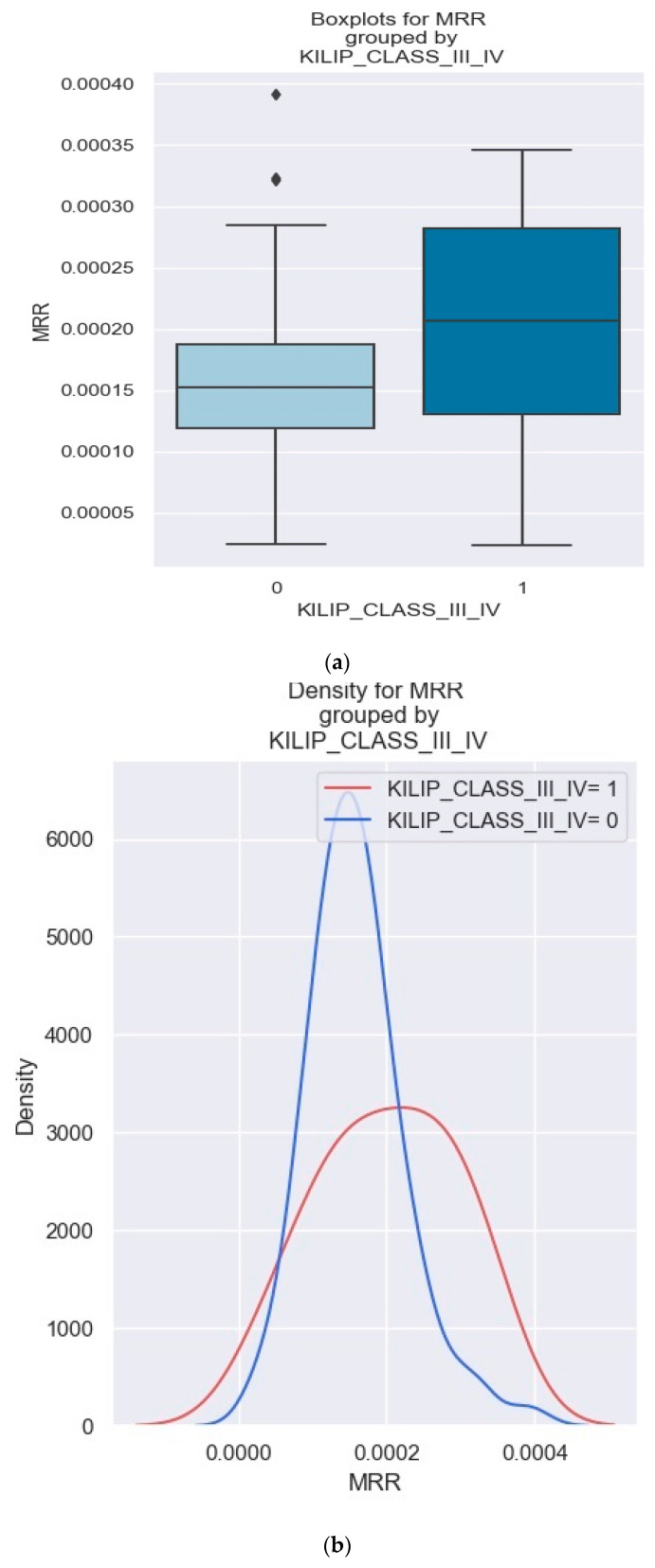
(**a**,**b**). Comparison of MRR levels between patients with and without Killip class III/IV. The figure is an original contribution by the authors.

**Figure 15 healthcare-12-00824-f015:**
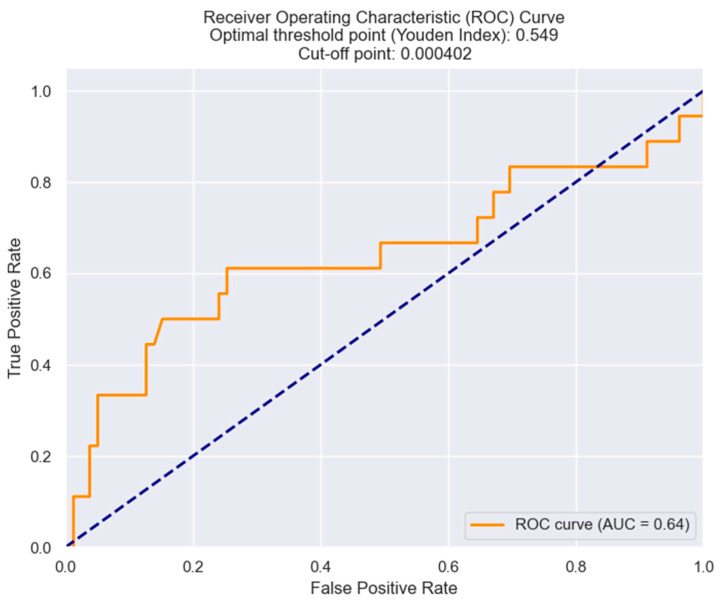
Receiver operating characteristic analysis and curve for predicting Killip class III/IV based on the MRR values. The figure is an original contribution by the authors.

**Figure 16 healthcare-12-00824-f016:**
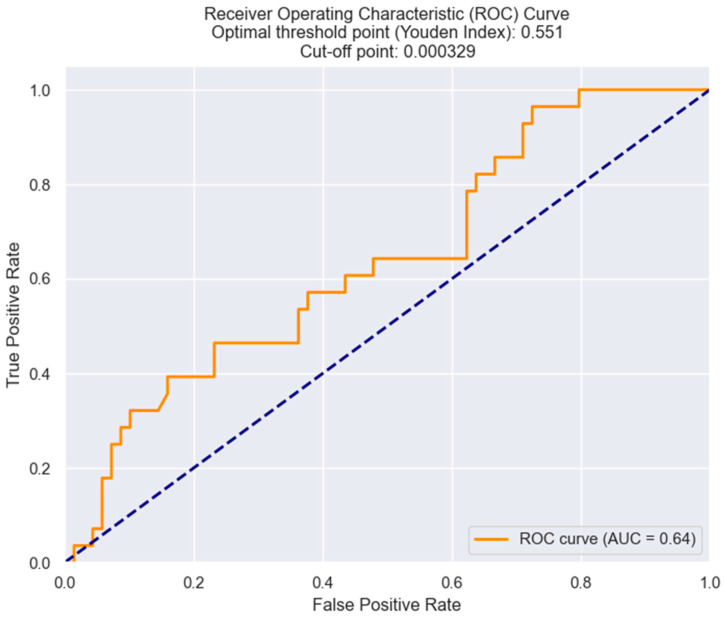
Receiver operating characteristic analysis and curve for predicting the necessity of surgical revascularization based on the MRR values. The figure is an original contribution by the authors.

**Table 1 healthcare-12-00824-t001:** Clinical and demographical characteristics of the study population.

Variable	Prevalence %
Female	36.08%
Male	63.92%
BMI >30	36.08%
Smoker	51.55%
Arterial Hypertension	79.38%
Diabetes mellitus	39.18%
Dyslipidemia	56.7%
Killip class III/IV	18.56%

The table is an original contribution by the authors.

**Table 2 healthcare-12-00824-t002:** Correlation between NRR and laboratory characteristics of NSTEMI patients.

Dependent Variable	R Squared	Independent Variable	Coefficient	*p* Value
NT-proBNP	0.099	NRR	1,952,000	0.008
Admission CK	0.05	NRR	117,000	0.028
Admission CK-MB	0.04	NRR	122,300	0.51
Admission high-sensitivity troponin I	0.025	NRR	1,751,000	0.123
Peak CK	0.019	NRR	122,000	0.182
Peak CK-MB	0.021	NRR	15,630	0.156
Peak high-sensitivity troponin I	0.086	NRR	5,659,000	0.004
Creatinine	0.198	NRR	219.1154	0.504

The table is an original contribution by the authors.

**Table 3 healthcare-12-00824-t003:** Correlation between NRR and clinical outcomes of NSTEMI patients.

Dependent Variable	R Squared	Independent Variable	Coefficient	*p* Value
Killip class III/IV	0.03933	NRR	658.3634	0.055
LVEF	0.040	NRR	−2722.8859	0.048
Triple vessel disease	0.0001099	NRR	34.5361	0.906
Indication for surgical revascularization	0.0346	NRR	612.9748	0.047
Length of hospital stay	0.028/	NRR	683.4844	0.103
In-hospital mortality	0.04906	NRR	81.2499	0.077
30-day readmission	0.009722	NRR	−412.5714	0.504
30-day mortality	0.002529	NRR	193.7938	0.748

The table is an original contribution by the authors.

**Table 4 healthcare-12-00824-t004:** Logistic regression applied to NSTEMI patients based on NRR value.

Dependent Variable	Killip Class III/IV	No. Observations	97			
Model	Logit	Df Residuals:	95.0			
Method	MLE	Df Model:	1.0			
Date	Tue. 3 October 2023	Pseudo R-squ.:	0.03933			
Converged	TRUE	LL-Null:	−46.534			
	Coef	Std err	Z	*p* > |z|	[0.025	0.975]
Intercept	−2.6494	0.696	−3.805	0.0	−4.014	−1.285
NRR	658.3634	342.773	1.921	0.055	−13.46	1330.187

The table is an original contribution by the authors.

**Table 5 healthcare-12-00824-t005:** Correlation between LRR and laboratory characteristics of NSTEMI patients.

Dependent Variable	R Squared	Independent Variable	Coefficient	*p* Value
NT-proBNP	0.099	LRR	−894,700	0.53
Admission CK	0.008	LRR	−79,710	0.386
Admission CK-MB	0.07	LRR	−8938	0.230
Admission high-sensitivity troponin I	0.015	LRR	−2,337,000	0.229
Peak CK	0.001	LRR	−45,080	0.774
Peak CK-MB	0.004	LRR	−10,930	0.563
Peak high-sensitivity troponin I	0.000	LRR	−510,200	0.881
Creatinine	0.000	LRR	11.444	0.895

The table is an original contribution by the authors.

**Table 6 healthcare-12-00824-t006:** Correlation between LRR and clinical outcomes of NSTEMI patients.

Dependent Variable	R Squared	Independent Variable	Coefficient	*p* Value
Killip class III/IV	0.04983	LRR	1203.1063	0.043
LVEF	0.00	LRR	−52.78	0.983
Triple vessel disease	0.0453	LRR	33.3808	0.947
Indication for surgical revascularization	0.001761	LRR	232.7326	0.647
Length of hospital stay	0.008	LRR	634.6635	0.379
In-hospital mortality	0.1264	LRR	865.9245	0.015
30-day Readmission	0.01596	LRR	660.7905	0.328
30-day mortality	0.0007319	LRR	198.672	0.87

The table is an original contribution by the authors.

**Table 7 healthcare-12-00824-t007:** Logistic regression applied to NSTEMI patients based on LRR value.

Dependent Variable	Killip Class III–IV	No. Observations	97			
Model	Logit	Df Residuals:	95.0			
Method	MLE	Df Model:	1.0			
Date	Tue. 3 October 2023	Pseudo R-squ.:	0.04983			
Converged	TRUE	LL-Null:	−46.534			
	Coef	Std err	Z	*p* > |z|	[0.025	0.975]
Intercept	−2.2424	0.477	−4.7	0.0	−3.178	−1.307
LRR	1203.1063	595.883	2.019	0.043	35.198	2371.015

The table is an original contribution by the authors.

**Table 8 healthcare-12-00824-t008:** Correlation between MRR and laboratory characteristics of NSTEMI patients.

Dependent Variable	R Squared	Independent Variable	Coefficient	*p* Value
NT-proBNP	0.067	MRR	15,390,000	0.032
Admission CK	0.000	MRR	70,260.0	0.896
Admission CK MB	0.002	MRR	2730	0.210
Admission high-sensitivity troponin I	0.015	MRR	−2,337,000	0.229
Peak CK	0.004	MRR	−584,900	0.521
Peak CKMB	0.010	MRR	−10,700	0.326
Peak high-sensitivity troponin I	0.002	MRR	7,873,000	0.697
Creatinine	0.104	MRR	1580.7824	0.001

The table is an original contribution by the authors.

**Table 9 healthcare-12-00824-t009:** Correlation between MRR and outcomes of NSTEMI patients.

Dependent Variable	R Squared	Independent Variable	Coefficient	*p* Value
Killip class III/IV	0.04963	MRR	7508.7325	0.033
LVEF	0.014	MRR	−16,280	0.244
Triple vessel disease	0.02149	MRR	4826.8294	0.104
Indication for surgical revascularization	0.04572	MRR	7143.8966	0.025
Length of hospitalization	0.001	MRR	1017.7005	0.809
In-hospital mortality	0.04044	MRR	1393.9474	0.105
30-day Readmission	0.0021	MRR	1717.5616	0.743
30-day mortality	0.01314	MRR	4302.8286	0.461

The table is an original contribution by the authors.

**Table 10 healthcare-12-00824-t010:** Logistic regression applied to NSTEMI patients based on MRR value.

Dependent Variable	Killip Class III–IV	No. Observations	97			
Model	Logit	Df Residuals:	95.0			
Method	MLE	Df Model:	1.0			
Date	Tue. 3 October 2023	Pseudo R-squ.:	0.04963			
Converged	TRUE	LL-Null:	−46.534			
	Coef	Std err	Z	*p* > |z|	[0.025	0.975]
Intercept	−2.8215	0.721	−3.911	0.0	−4.235	−1.408
MRR	7508.7325	3531.517	2.126	0.033	587.086	14,400.0

The table is an original contribution by the authors.

## Data Availability

The data presented in this study are available on request from the corresponding author. The data are not publicly available due to privacy issues.

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
