# Peer review of "Blood Cell Ratios Unveiled: Predictive Markers of Myocardial Infarction Prognosis"

_healthcare, 2024, doi:10.3390/healthcare12080824_

Round 1

Reviewer 1 Report

Comments and Suggestions for Authors

In my opinion, the bus analogy in the introduction is not appropriate for a scientific article.

The sample size (97 people) is very small even for an initial hypothesis test, especially given the availability of the method.

Publications twenty or more years ago, which are not unique, are unreasonably cited. There are more recent works on this topic.

The conclusions section needs revision.

I consider it inappropriate to refer to artificial intelligence, robotics and machine learning in conclusions, since this is not relevant to the research issues.

Reviewer 2 Report

Comments and Suggestions for Authors

Thank you for the opportunity to review this article. The topic of searching for predictors of the severity of myocardial infarction remains relevant to this day. I have several comments on the article: 1) The introduction does not disclose at all the scientific or practical significance of studying NRR, LRR or MRR indicators. What is the reason for the need to study these particular indicators if in the scientific literature there are many studies of the relationship between the level of red blood cell distribution width, neutrophil-to-lymphocyte ratio and the severity of myocardial infarction? 2) the style of writing the article should be more scientific; literary techniques are inappropriate here. 3) The sample of patients is too small to do reliable conclusions. The chapter "Materials and Methods" does not describe how the sample was determined. 4) Figure 1 would be better presented as a table. 5) there is no legend in Figure 2, the meaning is unclear. There are too many figures in the manuscript. There are inaccuracies in the text. For example, in line 361 the title of the table is “Correlation between LRR and laboratory characteristics of NSTEMI patients,” although the table talks about MRR.

Reviewer 3 Report

Comments and Suggestions for Authors

I had a pleasure of reviewing the manuscript entitled "The Players of the Bottom-Up Phenomenon in Non-ST-Elevation Myocardial Infarction”, which presents a comprehensive analysis of the potential role of neutrophil/red blood cell ratio (NRR), monocyte/red blood cell ratio (MRR), and lymphocyte/red blood cell ratio (LRR) as novel prognostic markers in patients with non-ST-elevation myocardial infarction (NSTEMI).
The subject is definitely of importance, especially since the primary goal of care in patients with acute coronary syndromes should be to identify patients at the highest risk of more dismal prognosis, and then to implement more effective strategies of mitigating that worse outcome.

Although the manuscript is interesting, there are several aspects which need to be raised. First of all, the peculiar manner of writing, including metaphorical presentation, and references to „brakes”, „bus”, or stating question marks, although such specific type of writing does not adhere to the highest standards of the peer-reviewed, highly revered journals. The introduction section of the manuscript is very little intuitive, and it requires a lot of focus and attention to understand the metaphores presented by the authors.

Furthermore, the limitations of the study are not adequately addressed. The Authors only briefly mention limitations related to the study population demographics and the challenges in blood sample acquisition, but a more comprehensive discussion of the study's limitations is necessary, since there are many more limitations, including the retrospective nature of the study. No details of the Registry have been published to date, and according to my knowledge, the Authors do not specify its registration in e.g. ClinicalTrials.gov or any other institutionalized database. No information is provided on the time frame of the study - considering that there were only 97 patients, it must have been conducted during approximately 1 calendar year, but such information must be presented. Was NSTEMI diagnosed on the basis of troponins, or both troponins and CKMB? If so, such information is important, since there might be major differences in clinical value of both measurements.
Considering inflammatory state, why was ESR, not CRP assessed, as the latter is a direct measure of inflammation?
Although the provided definition of renal dysfunction has been utilized in the past, the more prevalent, and recommended definition, and staging of CKD, is based on the other variables, such as eGFR (either calculated with MDRD or CKD-EPI formulas) or on the basis of ACR.
There is no information on how „triple” vessel disease was identified - were physiological exams, such as FFR, used for this assessment?
Moreover, no information on location of culprit lesion, periprocedural management, as well as pharmacotherapy, and detailed echocardiography, including valvular diseases are provided, what can significantly affect prognosis.

From the methodological point of view, I have some doubts concerning the analyses of the predictors of worse Killip Class on admission without such important parameters as time from symptom onset, LVEDD.

The title needs a major change, since based on reading the title, no reader would actually understand what the study is exactly about.

Finally, the mentioning of machine learning, or AI, in the conclusions, is somehow redundant.

Reviewer 4 Report

Comments and Suggestions for Authors

Comments on the Quality of English Language

minor English revision

Round 2

Reviewer 1 Report

Comments and Suggestions for Authors

Dear authors!

I have read the updated version of the article. Thank you for making changes based on my recommendations.

Reviewer 2 Report

Comments and Suggestions for Authors

The authors have significantly improved the quality of the manuscript. I believe that the article can be published as it is